# Dandelion (*Taraxacum mongolicum*) Extract Alleviated H_2_O_2_-Induced Oxidative Damage: The Underlying Mechanism Revealed by Metabolomics and Lipidomics

**DOI:** 10.3390/foods12173314

**Published:** 2023-09-03

**Authors:** Yannan Chen, Siyuan Fei, Xiaoting Yu, Mingqian Tan

**Affiliations:** 1Academy of Food Interdisciplinary Science, School of Food Science and Technology, Dalian Polytechnic University, Qinggongyuan, Gangjingzi District, Dalian 116034, China; ynchen04@163.com (Y.C.); 13470319363@163.com (S.F.); xiaoting12ming@126.com (X.Y.); 2National Engineering Research Center of Seafood, Dalian Polytechnic University, Dalian 116034, China

**Keywords:** dandelion, antioxidant, oxidative damage, metabolomics, lipidomics

## Abstract

Dandelion has received wide attention in food and medicine fields due to its excellent antioxidant properties. Nonetheless, the underlying mechanism of this action has not yet been fully clarified, particularly at the metabolic level. Herein, the effects of dandelion extract (DE) on H_2_O_2_-induced oxidative damage was investigated. The results indicate that the DE alleviated H_2_O_2_-induced cell damage (increased by 14.5% compared to H_2_O_2_ group), reduced the reactive oxygen species (ROS) level (decreased by 80.1% compared to H_2_O_2_ group), maintained the mitochondrial membrane potential (MMP) level, and increased antioxidant-related enzyme activities. Importantly, the metabolic response of PC12 cells indicates that H_2_O_2_ disturbed phospholipid metabolism and damaged cell membrane integrity. In addition, energy metabolism, the central nervous system, and the antioxidant-related metabolism pathway were perturbed. In contrast, DE rescued the H_2_O_2_-induced metabolic disorder and further alleviated oxidative damage. Collectively, these findings provide valuable stepping stones for a discussion of the mechanism and show the promise of DE as a suitable additive for functional food products.

## 1. Introduction

Oxidative stress is considered as one of the key factors related to human pathological processes, such as diabetes, chronic inflammation, cancers, and neurodegenerative diseases [1,2]. The abundance of reactive oxygen species (ROS) under oxidative stress leads to damage to cellular macromolecules, including protein modification, lipid oxidation, and DNA damage, which further causes the derangement of cellular functions and accelerates the occurrence of diseases [3]. In contrast, it is well known that a diet rich in antioxidants, especially the natural antioxidants, has a positive effect on maintaining the balance of ROS concentration and preventing a variety of pathophysiological processes [4]. Moreover, treatment based on supplements containing natural antioxidants lacks many of the side effects related to pharmacological treatment.

Dandelion (*Taraxacum mongolicum*), a member of the *Asteraceae* family, is widely used in traditional Chinese medicine and food supplements owing to its specific biological activities, such as anti-bacterial, anti-neoplastic, anti-inflammatory, and hepatoprotective properties, especially its antioxidant effect [5,6,7]. The different components from different organs of dandelion have demonstrated a strong anti-oxidative action [8,9]. For example, Jedrejek et al. found the phenolic fractions from the leaves of dandelion showed antioxidant activity against H_2_O_2_-induced oxidative damage [10]. Sun et al. confirmed dandelion aqueous extract could effectively alleviate LPS-induced oxidative stress in MAC-T cells [11]. Additionally, dandelion polysaccharides also showed significant antioxidant activity in various scavenging models [12]. Nevertheless, the current study of the effect of dandelion extract against oxidative damage is primarily focused on experimental investigations; the underlying mechanism of actions is not yet clear and needs to be discussed in depth.

Metabolomics is considered a powerful tool to detect changes in small molecular metabolites in organisms under different environmental stresses, which is helpful in investigating the underlying metabolic responses of the biological systems [13]. For instance, the level of fructose-6-phosphoric, pyruvate, and creatine in Raw264.7 cells changed significantly in the presence of H_2_O_2_, which further affected the amino acid and glucose metabolism pathways [14]. Thus, it is useful to understand the information about the metabolic profile variation, as well as the mechanism of H_2_O_2_-induced oxidative damage. However, the protective effects of natural antioxidants were scarcely investigated at the metabolic level.

Herein, the underlying mechanisms of DE against H_2_O_2_-induced oxidative damage were delineated by the combination of metabolomics and lipidomics. The effects of DE on H_2_O_2_-induced ROS accumulation, mitochondrial membrane potential variation, and antioxidant-related enzyme activities were investigated. Specially, the changes of metabolic pathways in the presence of H_2_O_2_ or DE-H_2_O_2_ were studied by the metabolomics and lipidomics method for the first time. This study might provide a comprehensive understanding of DE in alleviating H_2_O_2_-induced oxidative damage.

## 2. Materials and Methods

### 2.1. Materials

Dandelion was harvested in May 2022 at Tai’an, Shandong, China. These dandelions were identified as the dry whole plant of *Taraxacum mongolicum* by Professor Xiao Wang of Shandong University of Traditional Chinese Medicine. High glucose Dulbecco’s Modified Eagle’s medium (DMEM) was obtained from Hyclone Laboratories (Logan, UT, USA). 2′,7′-dichlorofluorescin diacetate (DCFH-DA), MitoTracker Red fluorescent probe, and Bisbenzimide H33342 trihydrochloride (Hoechst 33342) were supplied by Beyotime Biotechnology Co., Ltd. (Shanghai, China). Undecanoic acid, 19-decanoic acid were purchased from Aladdin Reagent Co., Ltd. (Shanghai, China). L-d_5_-phenylalanine was bought from Bai J&K Scientific Co., Ltd. (Beijing, China). Rat Adrenal Pheochromocytoma Cells (PC12 cells) were bought from Procell Life Science & Technology Co., Ltd. (Wuhan, China). Methanol and other chemical reagents were of chromatographic grade.

### 2.2. Preparation of DE

Dandelion was ground into powder by a disintegrator (800Y, Moling, Shanghai, China) and sieved with 60 mesh. Then, 10 g of dandelion powder was placed in a round-bottom flask and extracted upon refluxing twice with 60% ethanol (200 mL) for 2 h each time. The filtrates were combined and concentrated by a rotary evaporator (EYELAN-1100, Tokyo, Japan). The DE (1.6 g) was obtained by lyophilization.

### 2.3. Composition Analysis of DE

DE composition was characterized by liquid chromatography quadrupole time-of-flight/mass spectrometer (LC-Q-TOF/MS) analysis. Liquid chromatography was performed using the UHPLC system (H-Class, Waters, Milford, MA, USA). The samples were separated on an Acquity UPLC @BEH C8 column (100 mm × 2.1, 1.7 μm, Waters, Milford, MA, USA). The mobile phase A was water-formic acid (100:0.1, *v*/*v*) and B was acetonitrile-formic acid (100:0.1, *v*/*v*). The gradient program was optimized as follows: 0–5 min, 5–20% A; 10–15 min, 20–24% A; 15–25 min, 24% A; 25–26 min, 24–34% A; 26–35 min, 34–40% A, 35–50 min, 40–55% A, 50–60 min, 55–70% A, 60–70 min, 70–100% A. Tandem mass spectrometry (MS) was measured using a Q-TOF mass spectrometer equipped with an ESI interface (Impact II, Bruker, Karlsruhe, Germany). The data were acquired in positive and negative modes. The optimized parameters for the MS conditions were as follows: ion spray voltage, −3000 V (−)/3500 V (+); nebulizer pressure, 2.0 bar; flow rate of dry gas, 8.0 L/min; and drying gas temperature, 200 °C. The scan ranges in TOF-MS and TOF-MS/MS modes were both m/z 50–1200. For qualitative analysis, raw LC-Q-TOF/MS data were converted to ABF (analysis base file) format, and then processed using MS-DIAL 3.98. Through the comparison with self-built databases of metabolomics standards, along with analysis of the fragmentation pathways obtained from literature sources and public data, the compounds were characterized [15].

### 2.4. Cell Viability Assay

PC12 cells (1 × 10^5^ cells/well) were cultured in 96-well plates and treated with H_2_O_2_ (0, 100, 200, 400, 800, and 1000 μmol L^−1^), DE (0.00, 1.56, 3.12, 6.25, 12.50, 25.00, and 50.00 μg mL^−1^), or H_2_O_2_ (800 μmol L^−1^) in the presence of DE (12.50, 25.00, and 50.00 μg mL^−1^, named as low, medium, and high concentration) for 24 h. Afterwards, 20 μL of 3-(4,5)-dimethylthiahiazo(-z-y1)-3,5-di-phenytetrazoliumromide (MTT, 0.5 mg mL^−1^) was added to each well and cultured at 37 °C for another 4 h. After adding 150 μL DMSO to each well for 10 min, the absorbance value was detected at 570 nm using a microplate reader (Tecan, Hombrechtikon, Switzerland).

### 2.5. Hoechst 33342 Staining

PC12 cells (1 × 10^5^ cells/well) were seeded in 12-well plates and treated with different concentrations of DE for 6 h. After discarding the medium, the medium (1 mL) containing H_2_O_2_ (800 μmol L^−1^) was then added and stimulated for another 24 h. Finally, cells were stained by Hoechst 33342 for 10 min and imaged by fluorescent inverted microscope (Nikon Corp., Tokyo, Japan).

### 2.6. In Vitro Antioxidant Measurement

PC12 cells (1 × 10^5^ cells/well) were cultured in 12-well plates. Then, the cells were treated with different concentrations of DE for 6 h. After discarding the medium, the medium (1 mL) containing H_2_O_2_ (800 μmol L^−1^) was added and stimulated for another 24 h. ROS and MMP experiments were measured as previously reported [16]. Moreover, superoxide dismutase (SOD), glutathione peroxidase (GSH-Px), and malondialdehyde (MDA) were measured by the biochemical kit (Nanjing Jiancheng Bioengineering Institute, Nanjing, China).

### 2.7. Quantitative Real-Time PCR (RT-PCR) Measurement

The RT-PCR was measured by the following method of Chen et al., with minor modification [16]. Total RNA was obtained using TRIzol reagent, and the cDNA samples were obtained using a cDNA Synthesis Kit (Servicebio Technology Co., Ltd., Beijing, China). The relative mRNA expression levels were quantified by a real-time PCR system using SYBR Green qPCR Master Mix (Servicebio Technology Co., Ltd., Beijing, China), and normalized to GAPDH expression by the 2^−△△Ct^ method. The PCR primers information is listed in Appendix A.

### 2.8. Metabolomics and Lipidomics Analysis

PC12 cells were cultured with H_2_O_2_ (800 μmol L^−1^) or DE (50.00 μg mL^−1^)-H_2_O_2_ (800 μmol L^−1^) for 24 h. Later, the medium was removed and fresh medium was added and cultured for 2 h. Then, the precooled methanol (80% *v*/*v*, −80 °C) was added and incubated for 20 min at −80 °C. Undecanoic acid, 19-decanoic acid, and L-d_5_-phenylalanine were used as internal standards (5 μg/mL). After removing the cell fragment by centrifugation for 20 min (4 °C, 14,000× *g*), the supernatant was lyophilized. The solubilized metabolites were filtered through a 0.22 membrane before being analyzed. As for lipidomics analysis, lipids were extracted by methyl tert-butyl ether (MTBE) according to the published method with slight modifications. Phosphatidylcholine, phosphatidylethanolamine, and lysophosphatidylcholine were used as internal standards (5 μg/mL). Detailed conditions for chromatography and mass spectrometry of metabolomics and lipidomics were based on the previously reported methods [17] and are shown in the Appendix A.

### 2.9. Statistical Analysis

Dates were expressed as mean ± standard (SD). One-way ANOVA analysis was used to distinguish the statistical differences (*p* < 0.05). The multidimensional statistical analysis, including unsupervised principal component analysis (PCA) as well as supervised partial least squares discriminant analysis (PLS-DA), was performed through SIMCA 14.1 software. The heatmap was completed by TB tools 6.2.

## 3. Results and Discussion

### 3.1. LC-Q-TOF/MS Characterization of DE Composition

To reveal the underlying mechanism of DE against H_2_O_2_-induced oxidative damage, the DE composition was characterized by LC-Q-TOF/MS firstly (Appendix A). As displayed in Table 1 and Figure 1A, a total of 30 chemical components, including organic acids and flavonoids, were identified. These two types of substances are the main components that play an antioxidant role [4]. Organic acids contained caffeic acid, 1-O-caffeoylglycerol, p-hydroxycinnamic acid, protocatechuic acid, p-hydroxyphenylacetic acid, gallic acid, vanillic acid, chlorogenic acid, ferulic acid, tartaric acid, chicoric acid, isochlorogenic acid A, citric acid, caffeic acid-glucoside, shikimic acid, 3-O-feruloylquinic acid, quinic acid, and neochlorogenic acid, while flavonoids included rutin, quercetin, luteolin, isorhamnetin, isoquercitrin, luteolin-7-O-glucoside, apigenin, apigenin-7-O-glucoside, and naringin. Qu et al. also found similar results, identifying 28 organic acids and 19 flavonoids [18]. The difference in the number of compounds may be due to the different varieties of dandelion and different detection methods. Notably, the antioxidant activities of dandelion were mainly attributed to these organic acids and flavonoids [19].

### 3.2. Effects of DE on H_2_O_2_-Induced Cytotoxicity

The cytotoxicity evaluation was performed by MTT assay. As expected, the cell viability reduced with the increase in H_2_O_2_ (Figure 1B). Noteworthily, the cell viability decreased to 83% when the concentration of H_2_O_2_ was 800 μmol L^−1^ and exhibited a dramatic drop in the presence of 1000 μmol L^−1^ H_2_O_2_, indicating H_2_O_2_-induced severe oxidative damage to cells. Thus, 800 μmol L^−1^ H_2_O_2_ were selected for the further experiments. The cell viability showed negligible change under the addition of DE, which indicates good safety and biocompatibility (Figure 1C). As shown in Figure 1D, compared to the H_2_O_2_ group, the cell viability increased in the DE-H_2_O_2_ group, indicating that DE could alleviate H_2_O_2_-induced cytotoxicity. The cell viability was 95% in DE (50.0 μg mL^−1^)-H_2_O_2_ (800 μmol L^−1^) group, which was obviously higher than that of the H_2_O_2_ (800 μmol L^−1^) group (83%). Moreover, the obtained results were also confirmed by the morphology changes of PC12 cells (Figure 1E). Compared to the control group, the obvious change in cell morphology, i.e., abnormal sphericity, was observed after being treated with H_2_O_2_, and this phenomenon was alleviated in the presence of DE. The above results suggest that DE alleviated the cell cytotoxicity caused by H_2_O_2_.

### 3.3. Effects of DE on ROS Generation

Excessive generation of ROS could lead to cell apoptosis, so the concentration of ROS was measured to assess the intervention effects of DE against H_2_O_2_-induced oxidative damage [20]. DCFH-DA (an oxidation-indicative dye showing green fluorescence) was used to assess ROS production, and the produced green fluorescence intensity was positively related to the ROS level [21]. As shown in Figure 2, after treatment with H_2_O_2_, the green fluorescence intensity in PC12 cells increased to 224%, which indicated severe oxidative damage caused by H_2_O_2_. Compared to the H_2_O_2_ group, the green fluorescence intensity decreased after being pre-protected with different concentrations of DE, which decreased to 169%, 149%, and 123%, respectively. This indicates that DE alleviated the ROS generation and mitigated the occurrence of oxidative damage.

### 3.4. Effects of DE on MMP

Oxidative stress can cause mitochondrial dysfunction and reduce MMP. This is an early apoptotic event and can be measured by JC-1 [22]. JC-1 aggregates exhibit red fluorescence, suggesting higher MMP, while the JC-1 monomer with green fluorescence shows lower MMP. The fluorescence transition from red to green reflects the reduction in MMP [23]. As shown in Figure 3A, the H_2_O_2_ group induced an obvious reduction in MMP, as confirmed by the strongest green fluorescence and weakest red fluorescence. This finding was quantitatively displayed in the fluorescence intensity plots (Figure 3B,C). After addition of DE, the red/green fluorescence intensity ratio was obviously increased compared to the H_2_O_2_ group, especially at the high concentration of DE (Figure 3D). This indicates that DE alleviated the reduction of MMP caused by H_2_O_2_ and mitochondrial damage. The results were consistent with the tendency of the ROS analysis.

### 3.5. Effects of DE on Antioxidant-Related Enzyme Activities

Superoxide dismutase (SOD) and glutathione peroxidase (GSH-Px) are important antioxidant enzymes in human body, which can effectively scavenge excess oxygen-free radicals, and maintain cell vitality [24]. As shown in Figure 4A,B, the levels of SOD and GSH-Px reduced in H_2_O_2_ group. After the cells were pre-incubated with different concentrations of DE, the levels of SOD and GSH-Px increased in a dose-dependent manner. Meanwhile, the mRNA expression of SOD1 and GSH-Px showed the same trend (Figure 4C,D). Moreover, malondialdehyde (MDA), reflecting the degree of cell damage, was also detected by the assay kit. Compared to control group, the level of MDA increased significantly in H_2_O_2_ group, while decreased upon the addition of DE (Appendix A). Collectively, DE exhibited superior antioxidant abilities and were effective at alleviating H_2_O_2_ oxidative stress.

### 3.6. Effects of DE on Metabolic Response

The targeted metabolomics and lipidomics approaches were performed to evaluate the effects of DE on H_2_O_2_-induced oxidative damage at the metabolic level. The metabolic response of PC12 cells was evaluated after being treated with H_2_O_2_ (800 μmol L^−1^) and H_2_O_2_ (800 μmol L^−1^) in the presence of DE (50 μg mL^−1^) (DE-H_2_O_2_). Principal component analysis (PCA) score plots indicated that all QC samples were tightly clustered, suggesting the robustness of the capture method and the reliability of the data collected during the sample analysis (Appendix A). Next, PLS-DA was established to illustrate the different responses among the control, H_2_O_2_, and DE-H_2_O_2_ groups. As illustrated in Appendix A, the DE-H_2_O_2_ group presented an apparent separation from the control group and H_2_O_2_ group for metabolome and lipidomics, suggesting that H_2_O_2_ treatment significantly disrupted cellular metabolic processes, and that the addition of DE changed H_2_O_2_-metabolic disorders. Moreover, the model parameters (prediction ability (Q^2^) and credibility (R^2^)) for the metabolome were 0.757 and 0.901, and the Q^2^ and R^2^ for lipidomics were 0.809 and 0.987, respectively, suggesting satisfactory prediction and faithful representation. The permutation test was established to validate PLS-DA models (n = 200). It is shown in Appendix A that vector values of R^2^ and Q^2^ obtained in the permutation test were lower than the original value, indicating that the established model for the metabolome was stable and reliable. Similar results were also observed for lipidomics (Appendix A), which showed that the obtained vector values R^2^ and Q^2^ were smaller than the original values, indicating that no overfitting occurred in the PLS-DA score plot of lipidomics.

Figure 5 illustrates the change of the different metabolites and lipids after being treated with H_2_O_2_ or DE-H_2_O_2_. It was revealed that 36 metabolites were obviously changed in the H_2_O_2_ or DE-H_2_O_2_ group based on screening criteria (*p* < 0.05) (Appendix A), which were involved in glycolysis, the citric acid cycle (TCA cycle), antioxidant capacity, the central nervous system, energy metabolism, and amino acid metabolic pathways. Furthermore, enrichment analysis (based on *p* < 0.05) was used to study the effects of H_2_O_2_ or DE- H_2_O_2_ on various cellular metabolic pathways. Compared to the control group, H_2_O_2_ treatment significantly affected glycolysis, nicotinate and nicotinamide metabolism, serine and threonine metabolism, and phenylalanine, glycine, and pyruvate metabolism. After being pre-treated with DE, the disordered pathways induced by H_2_O_2_ were improved, and only phenylalanine, tyrosine and tryptophan biosynthesis were significantly disrupted (Appendix A). As for lipid metabolites, phosphatidylcholine (PC), phosphatidylethanolamine (PE), sphingomyelin (SM), fatty acids (FA), ceramides (Cer), triglycerides (TG), diglycerides (DG), hexosylceramides (HexCer), alkyl and alkenyl substituent PCs (PC-O), alkyl and alkenyl substituent PEs (PE-O) and cholesterol esters (CE) were significantly changed after different treatments (Figure 5B).

### 3.7. Effects of DE on Energy Metabolism

The results from metabolism analysis indicated that energy metabolism, including glycolysis, TCA cycle, nicotinate and nicotinamide metabolism, was obviously disturbed after intervention by H_2_O_2_ (Figure 6). For example, the intermediate metabolites (including glucose-6-phosphate, fructose-1,6-diphosphate, and pyruvic acid) were significantly changed. As one of the key limit-rate reactions in glycolysis, hexokinase catalysis can convert glucose into glucose-6-phosphate [25]. The relative content of glucose-6-phosphate in the H_2_O_2_ group was 4.8-fold higher than that in the control group, while it reduced by 14% in the presence of DE. Meanwhile, fructose 6-phosphate is catalyzed by phosphofructokinase to produce fructose 1,6-diphosphate, which is the second rate-limiting reaction in glycolysis [26]. The variation of the relative content of fructose-1,6-diphosphate exhibited a similar trend with glucose-6-phosphate after DE treatment. Moreover, as a direct carbon precursor for the synthesis of acetyl-CoA, pyruvate also exerts an essential role in the TCA cycle. The level of pyruvate significantly increased after treatment with H_2_O_2_, being 5.75-fold higher than that in control group. After being pre-protected with DE, the relative content of pyruvate decreased by 4.3% compared to H_2_O_2_ group. All these results suggest that glycolysis was perturbed when oxidative damage occurred and that DE could alleviate the disturbance of glucose metabolism induced by H_2_O_2_.

The TCA cycle participates in physiological processes, providing energy and building blocks for cells to sustain cell survival [27]. Compared to the control group, the level of oxaloacetate increased, indicating the disruption of TCA cycle after H_2_O_2_ addition. The level of oxaloacetate decreased after being treated with DE, which indicated the protective effect of DE against oxidative damage via the down-regulation of oxaloacetate [28]. Notably, nicotinate and nicotinamide metabolism were also disordered in the H_2_O_2_ group and reversed after being treated with DE. Nicotinate was metabolized to nicotinamide, a precursor of nicotinamide adenine dinucleotide (NAD+), which prevents oxidative damage by regulating energy metabolism [29]. All these results suggest that glycolysis, TCA cycle nicotinate and nicotinamide metabolism were perturbed when oxidative damage occurred and that DE could alleviate H_2_O_2_-induced metabolic disorder.

### 3.8. Effects of DE on Amino Acid Metabolism

Amino acids are fundamental constituents of proteins, which play an irreplaceable role in energy metabolism [30]. The amino acid metabolism was disrupted after treatment with H_2_O_2_ (Figure 6). The relative content of phenylalanine decreased by 26.58% compared with the control group. Reversely, the level of phenylalanine increased by 20.2% upon the addition of DE. As an indirect indicator of glycolysis, phenylalanine could influence the extracellular acidification rate (ECAR), which could reflect the activities of glycolysis [31]. These results demonstrate that DE could alleviate H_2_O_2_-induced glycolysis disorder. Moreover, the level of acetyl-lysine was up-regulated after being treated with H_2_O_2_, indicating the cells suffered oxidative damage. Acetyl-lysine was down-regulated to the normal level in the presence of DE, which was in line with the previous results [32].

### 3.9. Effects of DE on Other Metabolism Pathways

Studies have reported that tryptophan is involved in the regulation of oxidative stress, inflammation, and the immune response [33]. In addition, kynurenic acid is the downstream metabolite of tryptophan, which also plays an indispensable role in protecting the central nervous system [34]. As displayed in Figure 6, the levels of tryptophan and kynurenic acid were obviously changed after being treated with H_2_O_2_, indicating that the central nervous system was damaged. Reversely, the addition of DE mitigated the nervous system damage via participating in the metabolism pathway related to antioxidant. Bioactive compounds including creatine and its phosphate form (creatinine) were also changed after being treated with H_2_O_2_ and recovered to be almost equal to the control level in the presence of DE [35]. The above results demonstrate that oxidative stress occurred after being treated with H_2_O_2_ and was alleviated by the addition of DE.

### 3.10. Effects of DE on Lipid Metabolism

Extensive studies have identified bioactive compounds derived from fruit extracts and herbal products that could modulate lipid metabolism in a positive manner [36,37]. Thus, the effects of DE on H_2_O_2_-induced lipid metabolism disorders were elaborated (Figure 7). The transformation of SM to Cer is involved in cell apoptosis [38]. Compared to the control group, the level of SM was decreased by 32.3% and the level of Cer was increased by 70% after treatment with H_2_O_2_, indicating the apoptotic signal was active at these conditions. Phosphatides, including PC and PE, are important structural foundations of cell membranes. They participate in the cytidine diphosphate (CDP)-diacylglycerol and 1,2-diacylglycerol (DAG) pathways and are adjusted by various enzymes [39]. The PC content was increased after treatment with H_2_O_2_, illustrating that the cell membrane integrity was destroyed. Meanwhile, the intracellular glycerophosphatidylcholine also showed a change trend similar to that of PC. A previous study reported that glycerophosphatidylcholine is a metabolite of lipid oxidation and is also associated with the integrity of cell membranes [40]. Compared to the control group, the level of PE was decreased by 38.1% in the H_2_O_2_ group and was restored to the normal level in the presence of DE. LPC and LPE are the degradation products obtained by removing one molecule of fatty acid from PC and PE, respectively. In the oxidative stress state, the level of LPC and LPE increased, demonstrating that the catabolism of PC and PE was enhanced. The addition of DE alleviated the relative levels of PC and PE back to normal.

As an ether-containing phospholipid, plasmalogens play an important role in maintaining cell vitality and normal cell metabolism [41]. The disorder of plasmalogens is regarded as an indicator of oxidative stress. In the current study, the PC-O and PE-O were changed after incubation with H_2_O_2_, and the addition of DE was found to be able to alleviate this disorder (Appendix A). The change of plasmalogens was also found in As^3+^-induced oxidative stress, which further proved this finding [42].

Triglyceride (TG) is an important storage lipid in cells to load excessive fatty acid. TG accumulation caused by oxidative damage could be found in stressed cells [43]. Currently, the level of TG in the H_2_O_2_ group was 1.02-fold higher than in the control group. DG and FA are direct precursors of TG synthesis, which exhibited similar trends to TG under the same conditions [44]. After being treated with DE, the level of TG, DG, as well as FA decreased, implying that the addition of DE alleviated the H_2_O_2_-induced lipid metabolism disorder.

## 4. Conclusions

In summary, the underlying mechanism of DE against H_2_O_2_-induced oxidative damage was investigated at the metabolic level for the first time. Cell viability indicated that DE alleviated H_2_O_2_-induced cytotoxicity. In addition, DE efficiently protected PC12 cells against H_2_O_2_-induced oxidative damage by inhibiting ROS accumulation, increasing the MMP, and improving the antioxidant-related enzyme activities. Importantly, the presence of DE decreased the H_2_O_2_-induced energy metabolism, lipid metabolism, central nervous system and antioxidant-related metabolism disorder, thus alleviating oxidative damage. The results are helpful in clarifying the underlying mechanism of DE against H_2_O_2_-induced oxidative damage and provide new insights into dandelion as a suitable additive for functional food products.

## Figures and Tables

**Figure 1 foods-12-03314-f001:**
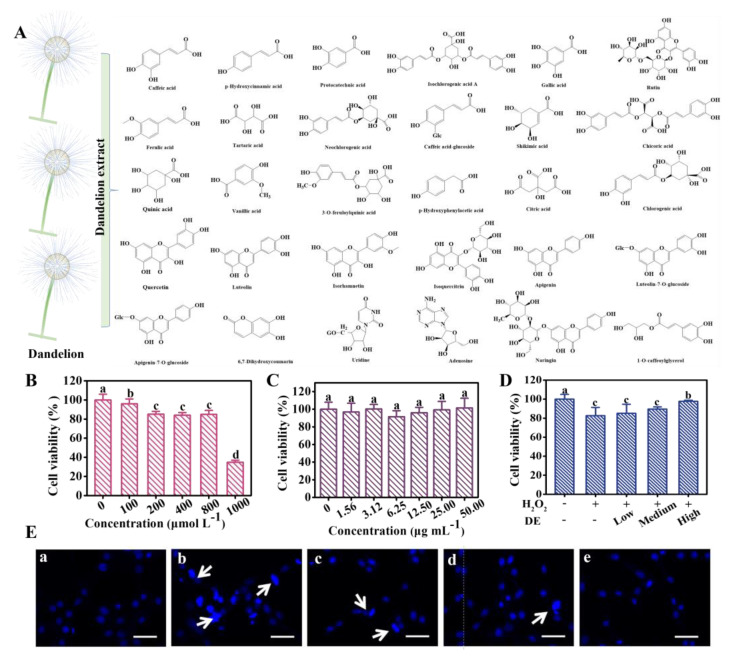
(**A**) Chemical structures of the compounds in dandelion extract. Cell viability of PC12 cells after incubation with (**B**) H_2_O_2_, (**C**) DE, and (**D**) H_2_O_2_ (800 μmol L^−1^) in presence of DE for 24 h. (**E**) Change in cell morphology after being treated with different groups: a, control group; b, H_2_O_2_ group; c, d, and e, H_2_O_2_ group in the presence of different concentrations of DE (12.5, 25 and 50 μg mL^−1^); arrows indicate the representative cells, scale bars were 100 μm. Different letters on the columns indicate significant differences (*p* < 0.05). Note: in (**D**), − indicates cells without any treatment, + indicates cells treated with 800 μmol L^−1^ H_2_O_2_.

**Figure 2 foods-12-03314-f002:**
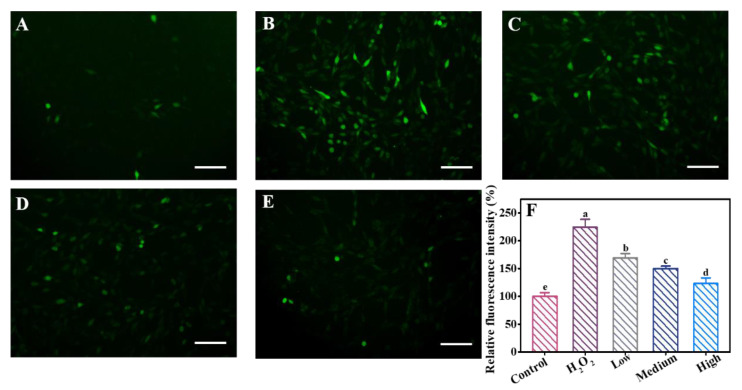
Fluorescence images of PC12 cells stained with DCFH-DA after the treatment of (**A**) DMEM medium (control), (**B**) H_2_O_2_, (**C**) Low concentration of DE+H_2_O_2_, (**D**) Medium concentration of DE+H_2_O_2_, and (**E**) High concentration of DE+H_2_O_2_. Scale bars were 100 μm. (**F**) Relative fluorescence intensity for different treatment groups. Different letters indicate significant difference at *p* < 0.05.

**Figure 3 foods-12-03314-f003:**
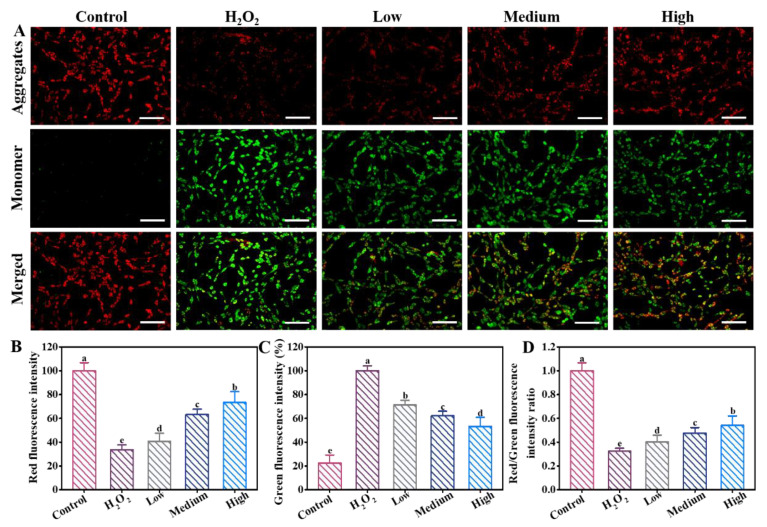
Fluorescence imaging of the mitochondrial membrane potential for (**A**) PC12 cells stained with JC-1 before (control) and after treatment with (H_2_O_2_), low, medium, and high concentration of DE. Scale bars were 100 μm. The fluorescence intensity of (**B**) red fluorescence and (**C**) green fluorescence. (**D**) Red/green fluorescence intensity ratio. Different letters indicate significant difference at *p* < 0.05.

**Figure 4 foods-12-03314-f004:**
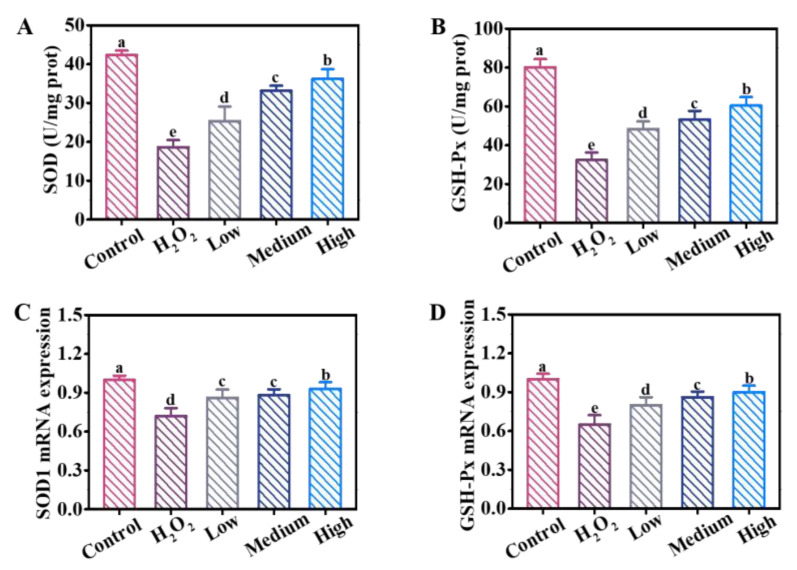
The level of (**A**) SOD and (**B**) GSH-Px in different treated groups. The mRNA expressions of (**C**) SOD1 and (**D**) GSH-Px in different treated groups. Different letters indicate significant difference at *p* < 0.05.

**Figure 5 foods-12-03314-f005:**
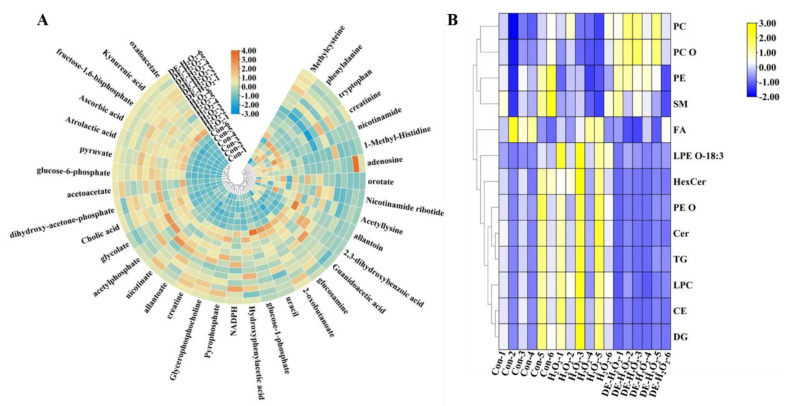
Heatmap of relative content of different metabolites for (**A**) metabolomics and lipids for (**B**) lipidomics analysis after being treated with H_2_O_2_ or DE-H_2_O_2_.

**Figure 6 foods-12-03314-f006:**
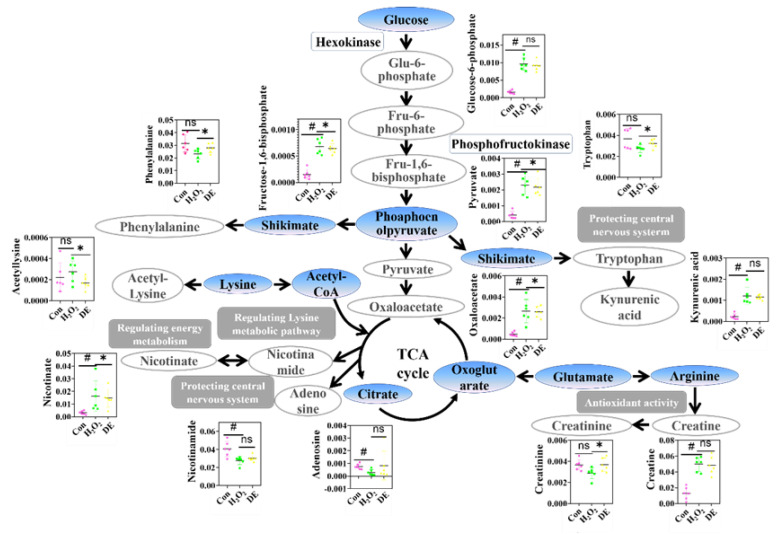
Changes of metabolic pathway after being treated with H_2_O_2_ or H_2_O_2_ in the presence of DE. “*” means significant differences (*p* < 0.05). “#” means significant differences (*p* < 0.01). “ns” means two columns have no difference.

**Figure 7 foods-12-03314-f007:**
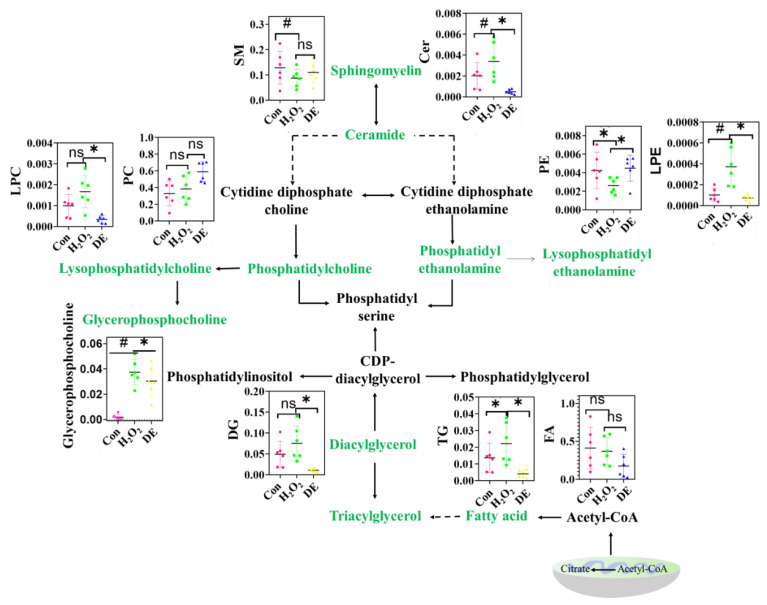
The change of lipid metabolic pathway after being treated with H_2_O_2_ or H_2_O_2_ in presence of DE. “*” means significant differences (*p* < 0.05). “#” means significant differences (*p* < 0.01). “ns” means two columns have no difference.

**Table 1 foods-12-03314-t001:** Identification of the components of dandelion extract.

No	Component	M^S^ (m/z, Da)	MS^2^ (m/z, Da)
1	Caffeic acid	179.0332[M-H]^−^	135.0357/117.0245
2	p-hydroxycinnamic acid	163.0288[M-H]^−^	119.0324/91.0246
3	Protocatechuic acid	153.0089[M-H]^−^	109.0047/91.0025
4	Isochlorogenic acid A	515.1178[M-H]^−^	353.0776
5	Gallic acid	169.0143[M-H]^−^	125.0124
6	Rutin	609.1449[M-H]^−^	301.0214/255.0103
7	Ferulic acid	193.0502[M-H]^−^	134.0203/115.0097
8	Tartaric acid	149.0092[M-H]^−^	87.0014
9	Neochlorogenic acid	353.0854[M-H]^−^	179.0247/93.0251
10	Caffeic acid-glucoside	341.0816[M-H]^−^	281.0649/179.0137
11	Shikimic acid	173.0421[M-H]^−^	110.8724
12	Chicoric acid	473.0642[M-H]^−^	219.0043/179.0247
13	Quinic acid	191.0558[M-H]^−^	93.0247
14	Vanillic acid	167.0345[M-H]^−^	152.0217/108.0125
15	3-O-Feruloylquinic acid	367.0124[M-H]^−^	193.0415/111.0386
16	p-hydroxyphenylacetic acid	151.0288[M-H]^−^	107.0124
17	Citric acid	191.0187[M-H]^−^	87.0018/67.0135
18	Chlorogenic acid	353.0834[M-H]^−^	191.0347/127.0298,
19	Quercetin	301.0324[M-H]^−^	150.9876/107.0032
20	Luteolin	285.0395[M-H]^−^	150.9941/133.0127,
21	Isorhamnetin	315.0506[M-H]^−^	300.0162/271.0034
22	Isoquercitrin	463.0864[M-H]^−^	301.0245/150.9934
23	Apigenin	269.0452[M-H]^−^	269.0353, 117.0264
24	Luteolin-7-O-glucoside	447.0928[M-H]^−^	285.0301
25	Apigenin-7-O-glucoside	431.0972[M+H]^+^	269.0357
26	6,7-Dihydroxycoumarin	177.0124[M-H]^−^	149.0132/133.0178
27	Uridine	243.0614[M+H]^+^	152.0237/110.0162
28	Adenosine	268.1024[M+H]^+^	136.0542
29	Naringin	581.1832[M+H]^+^	435.1224
30	1-O-caffeoylglycerol	253.0653[M-H]^−^	179.0102/133.0164

## Data Availability

The datasets generated for this study are available on request to the corresponding author.

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
