# Peer review of "Dandelion (Taraxacum mongolicum) Extract Alleviated H2O2-Induced Oxidative Damage: The Underlying Mechanism Revealed by Metabolomics and Lipidomics"

_foods, 2023, doi:10.3390/foods12173314_

Round 1

Reviewer 1 Report

Here's a corrected version of your text:

"In this study, the phytochemical analysis of Dandelion (Taraxacum mongolicum) extract and its efficacy in improving H2O2-induced oxidative damage investigated. Overall, the work is well-executed, and my suggestions are as follows:

In the abstract, it's crucial to include some data; readers should be able to discern key findings.

Compounds were identified using LC-QTOF mass spectrometry. I suggest including the MS spectra in the supporting information section. Furthermore, since you utilized the negative mode for analysis, there's no need to repeatedly mention [M-H]- in the table. Simply state it in the table's header."

This revised version maintains the original message while enhancing clarity and grammar.

Reviewer 2 Report

The paper of Yannan Chen et al. “Dandelion (Taraxacum mongolicum) extract alleviated H2O2-induced oxidative damage …” aimed to study of protection mechanism of Dandelion extract against H2O2-induced oxidative damage of cells. Generally, paper is well written and includes new scientific information.

Highlights and strengths of the manuscript are:

Chemical composition of Taraxacum mongolicum is well studied but protection mechanism of Dandelion extract against H2O2-induced oxidative damage of cells is under-researched.  The results may further increase interest in Taraxacum mongolicum application as medical plant.  

Specific comments and suggested revisions:

Table 1. It is not clear how authors identified metabolites; whether they used literature data or reference standards? Please, specify identification level for all metabolites (precise or tentative) accordingly commonly known classification (like 10.1007/s11306-007-0082-2).

Figure 1A  contains structure errors:

-          apigenin – it’s luteolin;

-          luteolin-7-glucuronide – not included in table 1; the wrong figure caption – luteolin-7-glucoside;

-          apigenin-7-glucoside – it’s apigenin;

-          naringin - does not understand the nature of monosaccharides link – naringin is neohesperidoside (2-O-rhamnosyl-glucose), narirutin is an isomer of naringin with rutinose (6-O-rhamnosyl-glucose) in 7-position;

-          structures of caffeoylquinic acids CQAs were a bit sloppy; the question is – why you cannot use general template to draw CQAs structures?

Section 3.2. It is not clear why did you choose H2O2 level 800 mcmol/L. Cell viability for 200, 400 mcmol/L were close to 800 mcmol/L (little over 80%). The best variant should be between 800 and 1000 mcmol/L. Exactly this influenced by the fact that there were minor differences in effectiveness of taraxacum extract groups.

All bioassays. There is, interestingly, though, a lack of standard positive control in all bioassays. Only blank, negative control and samples. How do you think I should estimate your results?

Supplementary. Tables and figures need to be captioned.

Reviewer 3 Report

This manuscript is well written and gives important information for the readers of Foods, however there are some problems and questions as follows.

1.       Authors should align the order of compounds names in Table 1 and each structure in Fig. 1A.

2.       Readers cannot know low, medium and high concentration of DE in Figs 1-5. Authors should show each concentration of DA concretely as low (12.5μg/mL), medium (25 μg/mL) and high (50μg/mL)? in the manuscript.

3.       Authors should insert “a – e” in Fig. 1E.

4.       Is it possible for authors to discuss between the content of bioactive compounds in DE and the results of metabolomics and/or metabolic pathway in Figs. 5-7?
